Assessing the living and dead proportions of cold-water coral colonies: implications for deep-water Marine Protected Area monitoring in a changing ocean

Vad Johanne jv63@hw.ac.uk 1 2
Orejas Covadonga 3
Moreno-Navas Juan 4
Findlay Helen S. 5
Roberts J. Murray Murray.Roberts@ed.ac.uk 2 6
1 School of Engineering Geoscience Infrastructure and Society, Heriot-Watt University , Edinburgh , United Kingdom
2 School of Geoscience, Grant Institute, University of Edinburgh , Edinburgh , United Kingdom
3 Instituto Español de Oceanografía, Centro Oceanográfico de Baleares , Palma , Spain
4 Physical Oceanography Research Group, Universidad de Málaga , Málaga , Spain
5 Plymouth Marine Laboratory , Plymouth , UK
6 Center for Marine Science, University of North Carolina at Wilmington , Wilmington , NC , United States of America
Vergés Adriana
Electronic publication date: 2017 Oct 5
Publication date: 2017
Volume: 5
Electronic Location ID: e3705
Received 2017 Mar 24; Accepted 2017 Jul 27
Copyright: ©2017 Vad et al.
Copyright year: 2017
Copyright holder: Vad et al.
License: This is an open access article distributed under the terms of the Creative Commons Attribution License, which permits unrestricted use, distribution, reproduction and adaptation in any medium and for any purpose provided that it is properly attributed. For attribution, the original author(s), title, publication source (PeerJ) and either DOI or URL of the article must be cited.
License URL: https://creativecommons.org/licenses/by/4.0/

Keywords: Cold-water corals, Lophelia pertusa, Mingulay Reef Complex, PISCES site, Rockall bank, Colony size, Dead framework, Ocean acidification, Deep-water marine protected areas, Monitoring

Funding: Natural Environment Research Council, the Department for Energy and Climate Change, and the Department for Environment, Food and Rural Affairs NE/H017305/1 European Commission’s H2020 scheme 678760 British Geological Survey Funding Initiative (BUFI) Natural Environmental Research Council NEM00578X/1 This paper is a contribution to the UK Ocean Acidification Research Programme (NE/H017305/1) to J. Murray Roberts; funded by the Natural Environment Research Council, the Department for Energy and Climate Change, and the Department for Environment, Food and Rural Affairs and the ATLAS project funded by the European Commission’s H2020 scheme through Grant Agreement 678760. Johanne Vad acknowledges support from the Natural Environment Research Council Centre for Doctoral Training in Oil & Gas (NEM00578X/1), through Heriot-Watt University (James Watt Scholarship scheme) and from the British Geological Survey (British University Funding Initiative). The funders had no role in study design, data collection and analysis, decision to publish, or preparation of the manuscript.

==============================
Coral growth patterns result from an interplay of coral biology and environmental conditions. In this study colony size and proportion of live and dead skeletons in the cold-water coral (CWC) Lophelia pertusa (Linnaeus, 1758) were measured using video footage from Remotely Operated Vehicle (ROV) transects conducted at the inshore Mingulay Reef Complex (MRC) and at the offshore PISCES site (Rockall Bank) in the NE Atlantic. The main goal of this paper was to explore the development of a simple method to quantify coral growth and its potential application as an assessment tool of the health of these remote habitats. Eighteen colonies were selected and whole colony and dead/living layer size were measured. Live to dead layer ratios for each colony were then determined and analysed. The age of each colony was estimated using previously published data. Our paper shows that: (1) two distinct morphotypes can be described: at the MRC, colonies displayed a ‘cauliflower-shaped’ morphotype whereas at the PISCES site, colonies presented a more flattened ‘bush-shaped’ morphotype; (2) living layer size was positively correlated with whole colony size; (3) live to dead layer ratio was negatively correlated to whole colony size; (4) live to dead layer ratio never exceeded 0.27. These results suggest that as a colony develops and its growth rate slows down, the proportion of living polyps in the colony decreases. Furthermore, at least 73% of L. pertusa colonies are composed of exposed dead coral skeleton, vulnerable to ocean acidification and the associated shallowing of the aragonite saturation horizon, with significant implications for future deep-sea reef framework integrity. The clear visual contrast between white/pale living and grey/dark dead portions of the colonies also gives a new way by which they can be visually monitored over time. The increased use of marine autonomous survey vehicles offers an important new platform from which such a surveying technique could be applied to monitor deep-water marine protected areas in the future.

Introduction

Some species of cold-water corals (CWC) can form complex 3-dimensional reef frameworks supporting biodiversity hotspots (e.g., Freiwald, 2002; Roberts et al., 2009; Henry, Davies & Roberts, 2010; Buhl-Mortensen et al., 2010), but several anthropogenic activities are putting them at risk. Fisheries, oil and gas extraction, deep-sea mining as well as the effects of climate change, including ocean acidification (OA), are threatening these important benthic communities (e.g., Koslow et al., 2000; Roberts, Wheeler & Freiwald, 2006; Hall-Spencer et al., 2008; Hennige et al., 2015; Büscher, Form & Riebesell, 2017). One of the factors that clearly defines the resilience of such fragile benthic communities to natural and anthropogenic impacts, as well as the population dynamics of clonal species such as corals, is their growth rate and growth pattern (Hughes, 1987). However, in comparison with their tropical counterparts, azooxanthellate CWC still remain less known and much less studied due to the difficulties in accessing their remote deep-sea locations. Coral growth is controlled by a range of environmental factors. In deep waters (>100 m), local hydrodynamics and energy supply (Mienis et al., 2007), as well as temperature (Thresher, 2009) play a central role for survival and growth of CWC. For many years, these factors have also been known to modify tropical coral phenotype. For example, branching tropical corals tend to become less robust with depth (Barnes, 1973) but less is known about the influence of environmental factors on CWC phenotypes. Over the last 15 years improvements in aquaria facilities (e.g., Roberts & Anderson, 2002; Olariaga et al., 2009) and in the growing use of high resolution visual surveys from Remotely Operated Vehicles (ROV), have allowed to advance our understanding of these previously unreachable ecosystems. Video footage and still images obtained with ROVs have become powerful non-destructive approaches to study several aspects of CWCs and the communities they support including their occurrence, density and geographic distribution (e.g., Orejas et al., 2009; Arnaud-Haond et al., 2015), bathymetric distribution, coral size classes and orientation (Gori et al., 2013) and relationship with associated species (Purser et al., 2013).

Our knowledge of CWC growth rates has also dramatically improved thanks to experimental studies (e.g., Orejas, Gori & Gili, 2008; Orejas et al., 2011a; Orejas et al., 2011b; Brooke & Young, 2009; Lartaud et al., 2013) and field measurements on man-made structures (Gass & Roberts, 2006; Gass & Roberts, 2011; Larcom et al., 2014). Moreover, data both on abiotic parameters and from video and photographic records can now allow morphological patterns and colony biometrics to be described, quantified and related to abiotic environmental data. This synergistic approach linking colony morphology and size to environmental parameters is needed to gain deeper understanding of the relationship between those drivers and CWC growth. These are critical steps necessary if field monitoring programmes are to be established to understand and record the implications of global change on CWC habitats.

To date most laboratory CWC studies have worked on Lophelia pertusa (Linnaeus, 1758). Several studies show the degree of adaptation of this species to temperature changes (e.g., Dodds et al., 2007; Brooke et al., 2013; Naumann, Orejas & Ferrier-Pagès, 2014). In recent years, much effort has been made to understand the effects of OA on L. pertusa growth (e.g., Form & Riebesell, 2011; McCulloch et al., 2012; Maier et al., 2013; Movilla et al., 2014; Hennige et al., 2015) and the carbonate chemistry of the environments where it occurs (e.g., Findlay et al., 2014). The recent results from studies performed using L. pertusa from the Mingulay Reef Complex (Roberts et al., 2005; Roberts et al., 2009) indicated that this species was fairly resilient under OA scenarios under short timescales of 21 to 89 days (Maier et al., 2013; Hennige et al., 2014b). However, over longer experimental time periods of a year, biomineralisation processes in L. pertusa changed inducing modifications in polyp morphology and making the skeleton more fragile, with additional evidence that dead portions of the skeleton not covered by living coral tissue were particularly vulnerable (Hennige et al., 2015). Thus, quantifying the proportions of live and dead coral during field surveys is an essential prerequisite of any long-term monitoring programme to follow CWC framework reefs over time.

In this study, we explore a new approach to assess L. pertusa colony size, and the proportion of live and dead coral in each colony by using opportunistic measurements from high definition video footage recorded from two sites in the NE Atlantic, one inshore (the Mingulay Reef complex) and one offshore (the PISCES site). We hypothesize that proportion of live and dead coral in L. pertusa colonies will differ between and within sites as abiotic conditions change. This opportunistic study was completed with the footage available from the 2012 Changing Oceans Expedition (RRS James Cook cruise 073) and revealed: (1) that distinct colony morphotypes dominate each study site and (2) that both morphotypes were predominantly composed of dead coral with smaller proportions of live coral polyps found in all colonies analysed. Based on these preliminary results, we explore the potential applications of this coral growth quantification as an assessment tool to determine the health and conservation status of deep-sea framework building corals.

Figure 1 (A) Mingulay Reef Complex (MRC) and PISCES area location offshore Scotland. (B) Colonies 1 to 14 (C1 to C14) locations within MRC. Note that colony 5 (C5) is the only colony located on Banana Reef. (C) Colonies 15 to 18 (C15 to C18) locations within PISCES area.

Material & Methods

Research area

The Mingulay Reef Complex (MRC) is located in the Sea of the Hebrides between the uninhabited island of Mingulay and the west coast of Scotland (Fig. 1). Within the MRC, Mingulay Reef Area 1 (MR) constitutes two asymmetric east–west oriented ridges 1.5 and 2.3 km long respectively separated by an approximately 700 m wide gap (Fig. 1). The so-called Banana Reef (BR) to the southeast of MR is formed by a thin 2.5 km-long ridge (Fig. 1) (Roberts et al., 2009; Duineveld et al., 2012). The coral colonies forming the MRC grow preferentially on the topographic highs created by the flanks and crests of ridges formed by dolorite sills (MR) and igneous intrusions (BR) that outcrop at the seabed (Roberts et al., 2009). Video transects used in this study revealed a high coral cover but our analyses were at times constrained by poor visibility at this site in the post-spring bloom time period when the surveys were carried (RRS James Cook cruise 073, Roberts & shipboard party, 2013).

Comprehensive information on abiotic factors were available throughout MRC: depth and site carbonate chemistry were known thanks to successive surveys starting in 2003 (Roberts et al., 2005; Roberts et al., 2009; Davies et al., 2009; Findlay et al., 2013). Furthermore average current speed, current speed standard deviation, maximum current speed, aspect, slope as well as rugosity were extracted from a high resolution 3D hydrodynamic model with 100 m spatial resolution developed at constant values for temperature and salinity (Moreno-Navas et al., 2014). These seabed terrain variables were calculated with a spatial resolution of 3 m using ArcGis 9.2 with ESRI spatial analysis extension (Moreno-Navas et al., 2014).

The other study location was the PISCES site on the Rockall Bank, 460 km west of Scotland. The offshore PISCES site was first described by Wilson (1979) during research submersible dives using PISCES III in 1973. However, it has been less intensively studied than the MRC. In this area, L. pertusa shows a discontinuous patchy distribution of ‘Wilson rings’, mostly at depths of 220–350 m where coral colonies grow on the flanks of Pleistocene iceberg ploughmarks (Wilson, 1979). Video transects recorded here also covered extensive off-reef habitats, illustrating the sparsity of the coral cover at the PISCES site in comparison with MRC (RRS James Cook cruise 073, Roberts & shipboard party, 2013). The hydrodynamic model available for MRC do not extend to the PISCES site and abiotic factors values for colonies for the PISCES site were therefore not available.

Sampling and video processing

In the present study, video surveys were conducted at MRC and PISCES, during the 2012 Changing Oceans Expedition (RRS James Cook cruise 073, Roberts & shipboard party, 2013) carried out through the Natural Environment Research Council’s UK Ocean Acidification research program (UKOA, NERC). The cruise took place in May–June 2012 and high definition video footage was recorded with the Holland-1 ROV, from the Irish Marine Institute (Galway). ROV position was recorded by an ultra-short-baseline (USBL) underwater positioning system.

A total of nine video surveys, seven from MRC (six from MR and one from BR) and two from PISCES were used in this study. From these dives, all colonies (1) visibly distinctive from others on the video footage, (2) close enough to the ROV for precise measurements (ROV within 1 m of the colony) and (3) displaying a clearly visible separation between the darker dead and the brighter white living layers of the coral colony were selected. Still images of those colonies were extracted from the videos (Figs. 2 and 3). In this study, a L. pertusa colony refers to a distinctive coral sub-entity of the reef. We recognise that skeletal fusion in L. pertusa is common (Hennige et al., 2014a) and therefore do not use the word colony to imply any genetic differentiation.

Figure 2 Method applied to measure living and dead layers of a colony at MRC.

(A) Original capture from the video record. (B) Schematic image of the colony overlaying original image. Colours black, brown and red display different polyp layers from the front part to the back of the colony, the green colour displays the groups of polyps which are in the same plane chosen here to measure the dead and the living layer. (C) Schematic image of the colony with measure bars. (D) Original image with measure bars.

Figure 3 Method applied to measure living and dead layers of a colony at Pisces.

(A) Original capture from the video record. (B) Schematic image of the colony overlaying original image. Colours black, brown and red display different polyp layers from the front part to the back of the colony, the green colour displays the groups of polyps which are in the same plane chosen here to measure the dead and the living layer. (C) Schematic image of the colony with measure bars. (D) Original image with measure bars.

Universal time codes were not embedded in the high definition footage but were available via low definition images recorded by three additional ROV cameras. Synchronisation of the two video records enabled the time each colony was filmed to be extracted and from this their precise positions (Universal Transverse Mercator) were logged using the Ocean Floor Observation Protocol (OFOP) (Huetten & Greinert, 2008) navigation system output. Videos were replayed and processed using iMovie (Apple Inc.).

Colony measurements and age estimation

The Holland-1 ROV was equipped with two laser scale pointers separated by 100 mm which were used to assess overall coral colony size and the thicknesses of the dead and living layers within each colony. On rare occasions, the two laser beams were not visible because of high water turbidity. In these cases, it was possible to estimate colony sizes using the known dimensions (400 mm width) of a bio-box sampling unit mounted immediately adjacent to the corals in the field of view at the front of the ROV (see Fig. 2A). To limit perspective errors only colonies immediately adjacent to the bio-box were measured in this way.

Each colony image was processed using the free software ImageJ (Rasband, W.S., ImageJ, US National Institutes of Health, Bethesda, Maryland, USA, http://imagej.nih.gov/ij/, 1997–2014). For each image, the size of the whole colony (from base to top) and of each layer (dead and living part) was measured at five different points on each colony in order to best catch intra-individual variability. To perform these measurements, the growth direction of the coral branches was followed but took into account the three dimensional nature of the colonies as shown in Figs. 2 and 3. Thus, after image processing, five measurements per colony were available for the whole colony size as well as for the dead and the living layer (Figs. 2 and 3). The ‘living layer thickness: whole colony size’ ratio (LL : WC ratio) was also calculated for each colony dividing the thickness of the living layer by the size of the whole colony.

Age for each colony was determined using a growth rate estimation of 26 ± 5 mm yr−1 (Gass & Roberts, 2006). Although several studies have assessed different L. pertusa growth rate using in-situ measurement, coral staining and aquaria approaches (Duncan, 1877; Dons, 1944; Orejas, Gori & Gili, 2008; Orejas et al., 2011b; Brooke & Young, 2009; Lartaud et al., 2013), the Gass & Roberts (2006) estimation is based on in situ colony observation over time in the North Sea using ROV recorded video footage and therefore gives the closest match geographically and in terms of water depth to this study.

Numerical and statistical analysis

Average values of layer thicknesses, whole colony sizes and LL : WC ratios were determined for each colony.

To determine differences in measurements between colonies across sites, two-sided two samples Wilcoxon tests were performed for each metric. Further statistical analysis was also carried out to determine Spearman correlation factors and p-values between the whole colony size and the living layer thickness as well as the LL : WC ratio for each colony. For these calculations, all of the measurements were used (18 colonies × 5 replicates for each colony).

To determine the influence of abiotic factors on CWC growth, average current speed, current speed standard deviation, current speed maximum as well as depth and aspect (facing gradient of the seabed), slope (gradient) and rugosity were extracted at each MRC colony location from the model described by Moreno-Navas et al. (2014). This model was not available for the PISCES site colonies, it was developed only for MRC with constant temperature and salinity (Moreno-Navas et al., 2014). Spearman correlation coefficient and p-value were respectively calculated between these abiotic factors and each layer size as well as -log transformed ratio. All statistical handling was performed with the free software R (R Core Team, 2013).

Results

Lophelia pertusa colonies morphotypes from MRC and PISCES

In total 18 colonies from nine transects varying from 63 to 1,865 m in length, displayed a clear live/dead layer separation and could thus be used in the analysis (Table 1, Fig. 1). Fourteen of these colonies were located at MRC (colonies 1 to 14) and the four remaining were situated in PISCES (colonies 15 to 18) (Table 1).

Table 1 Transect number, location, coordinates (latitude longitude) at the start and the end of each transect, depth (m) at the start and the end of the transect, length (m) and number of selected colonies analysed in this study.

Transect	Location	Transect coordinates	Transect length (m)	Start depth (m)	End depth (m)	Number of selected colonies	
		From	To					
1	MR	56°N 49.61	7°W 23.39	56°N 49.52	7°W 23.48	250	157	179	1	
3	56°N 49.38	7°W 23.71	56°N 57.59	7°W 13.04	63	133	127	2	
5	56°N 82.27	7°W 39.51	56°N 82.29	7°W 39.48	152	133	130	1	
8	56°N 49.59	7°W 22.21	56°N 49.29	7°W 22.85	1,236	167	130	6	
10	56°N 49.36	7°W 23.69	56°N 49.38	7°W 23.68	84	130	127	2	
41	56°N 49.55	7°W 23.49	56°N 49.43	7°W 23.3	238	150	127	1	
7	BR	56°N 48.13	7°W 27.01	56°N 48.39	7°W 25.98	1,226	145	155	1	
31	PISCES	57°N 61.01	14°W 49.25	57°N 60.58	14°W 49.67	1,865	262	260	2	
32	57°N 59.49	14°W 51.27	57°N 59.49	14°W 50.96	613	262	261	2	
									(Total = 18)	
Notes.

MR Mingulay Reef Area 1

BR Banana Reef

Differences in colony morphology were identified between the two areas: L. pertusa colonies at MRC displayed a spherical “cauliflower” shape (sensu Freiwald, Wilson & Henrich, 1999; Rogers, 2004; Orejas et al., 2009), resulting from a multidirectional growth (Fig. 2). On the contrary, L. pertusa colonies at PISCES were less abundant than in MRC, flattened and horizontally planar in shape, emerging from a horizontal growth (Fig. 3). Wilson (1979) called the PISCES morphotype ‘bush-shaped’ and these colonies displayed a less compact shape than those at MRC with some portions of the colonies not covered by living polyps (Fig. 3).

Colony size and layer thickness estimation

Overall L. pertusa whole colony size ranged from 324 ± 43 mm (Colony 2, MRC) to 1,344 ± 115 mm (Colony 13, MRC) (Table 2, Fig. 4). Therefore, the estimated ages of the colonies observed in this study ranged from 12.9 ± 3.1 (colony 2, MRC) to 53.7 ± 11.7 years old (colony 13, MRC) (Table 2).

Figure 4 Lophelia pertusa (A) number of colonies belonging to the different size ranges detected in the Mingulay Reef Complex (MRC) and the PISCES area and (B) Living (grey) and dead (antracit) layer sizes for the 20 L. pertusa colonies analysed (14 from MRC, including colony 5 which has been recorded in Banana Reef; 5 from PISCES).

Error bars display the SD.

Table 2 Lophelia pertusa colony layer and whole size measurements (mm ± SD) and Living layer: Whole size ratio estimation (LL : WC) and age estimation based on Gass & Roberts (2006).

Area	Colony number	Dead layer (mm)	Living layer (mm)	Total colony size (mm)	Living layer: whole size ratio	Age estimation (years)	
MRC	1 (1)	747 ± 66	126 ± 30	873 ± 90	0.14 ± 0.02	34.9 ± 7.9	
2 (3)	276 ± 32	48 ± 13	324 ± 43	0.15 ± 0.02	12.9 ± 3.1	
3 (3)	498 ± 57	136 ± 7	634 ± 64	0.22 ± 0.01	25.3 ± 5.7	
4 (5)	692 ± 25	176 ± 16	868 ± 32	0.20 ± 0.02	34.7 ± 7.1	
5a (7)	719 ± 75	260 ± 10	979 ± 75	0.27 ± 0.02	39.1 ± 8.4	
6 (8)	584 ± 21	76 ± 13	660 ± 23	0.12 ± 0.02	26.4 ± 5.4	
7 (8)	463 ± 52	129 ± 19	592 ± 37	0.22 ± 0.05	23.7 ± 5.0	
8 (8)	997 ± 73	207 ± 26	1,204 ± 87	0.17 ± 0.02	48.1 ± 10.3	
9 (8)	940 ± 78	110 ± 7	1,051 ± 79	0.11 ± 0.01	42.0 ± 9.0	
10 (8)	1,045 ± 58	204 ± 17	1,249 ± 66	0.16 ± 0.01	49.9 ± 10.4	
11 (8)	675 ± 89	160 ± 28	835 ± 104	0.19 ± 0.03	33.3 ± 7.8	
12 (10)	1,070 ± 92	196 ± 20	1,265 ± 95	0.16 ± 0.02	50.5 ± 10.9	
13 (10)	1,208 ± 124	136 ± 18	1,344 ± 115	0.10 ± 0.02	53.7 ± 11.7	
14 (41)	268 ± 9	76 ± 10	344 ± 19	0.22 ±0.02	13.7 ± 2.9	
PISCES	15 (31)	964 ± 200	148 ± 33	1,112 ± 226	0.13 ± 0.02	44.4 ± 12.5	
16 (31)	780 ± 82	165 ± 19	945 ± 70	0.18 ± 0.03	37.7 ± 8.1	
17 (32)	946 ± 181	181 ± 38	1,057 ± 155	0.17 ± 0.03	45.0 ± 10.9	
18 (32)	828 ± 53	180 ± 32	1,010 ± 71	0.18 ± 0.03	40.3 ± 8.6	
Notes.

a Colony 5 is the only colony present at Banana Reef. Transect number is displayed between brackets in the second column.

The living layer size stayed relatively stable: varying from a minimum of 48 ± 13 mm (colony 2, MRC) to a maximum of 260 ± 10 mm (colony 5, MRC) (Table 2, Fig. 4). In contrast to the living layer, the dead layer thickness varied notably between colonies, accounting for the variability of the whole colony size described above. Colony 13 showed the largest dead layer (1,208 ± 124 mm) whereas the smallest dead layer (207 ± 26 mm) was measured in colony 9 (Table 2, Fig. 4).

Difference in layer sizes and whole colony sizes between sites was not observed. However, living and dead layer thickness variations were overall less notable at the PISCES site (Table 2, Fig. 4).

In all the colonies measured in this study, the living layer never exceeded one fourth of the whole colony size, resulting in LL : WC ratios ranging from 0.10 ± 0.02 (colony 13, MRC) to 0.27 ± 0.02 (colony 5, MRC) (Table 2). LL : WC ratio variation for colonies from PISCES was very narrow with minimum values of 0.16 ± 0.02 (colony 16) to maximal of 0.18 ± 0.03 (colony 15 and 18) (Table 2).

Statistical analysis

All two-sided two samples Wilcoxon tests (on layer thicknesses, whole colony sizes and -Log transformed LL : WC ratio) produced non-significant p-values (ranging from 0.1515 to 0.915) showing no statistically significant differences between the colonies across sites (Table 3) for all the metrics measured here.

Spearman correlation tests however revealed a significant positive correlation (p-value = 1.1e−11, ρ = 0.63) between whole colony size and living layer size and a significant negative correlation (p-value = 4.4e−04, ρ =  − 0.35) between whole colony size and LL : WC ratio (Fig. 5).

Table 3 Two sided two samples Wilcoxon test comparing mean dead layer thickness, mean living layer thickness, mean whole colony size and mean LL : WC ratio between MRC and Pisces sites.

	Two-sample Wilcoxon test	
	Wilcoxon W	p-Value	
Dead layer thickness	14	0.1515	
Living layer thickness	15	0.1837	
Whole colony size	19	0.3817	
LL : WC ratio	29.5	0.915	

Figure 5 Scatterplot of living layer thickness (A) and LL : WC ratio (B) as a function of total colony sizes.

Measurements from MRC colonies are displayed in black; measurements from PISCES colonies are displayed in grey. Regression lines for each plot are traced in green.

Correlation coefficients between living and dead layer thicknesses and abiotic factors available at MRC (depth, average current speed, aspect, slope and rugosity) (Table 4) were overall low and not significant as all p-values varied between 0.36 and 0.98.

Table 4 Depth (m), average current speed (m/s), current speed standard deviation (m/s), maximum current speed (m/s), aspect (°), slope (°) and rugosity (µm) values extracted from Moreno-Navas et al.’s (2014) hydrodynamic model at each MRC colony location.

Colony number	Depth (m)	Average current velocity (m/s)	Standard deviation current velocity (m/s)	Maximum average current velocity (m/s)	Aspect (∘)	Slope (∘)	Rugosity (μm)	
1 (1)	173.28	0.277	0.126	0.527	19.854	6.602	1.010	
2 (3)	132.78	0.340	0.155	0.636	30.360	7.250	1.020	
3 (3)	126.89	0.336	0.157	0.637	334.868	9.826	1.185	
4 (5)	131.02	0.336	0.157	0.637	247.241	12.998	1.049	
5 (7)	143.86	0.296	0.074	0.450	81.671	0.551	1.001	
6 (8)	146.16	0.291	0.150	0.548	325.656	13.823	1.061	
7 (8)	141.16	0.291	0.150	0.548	319.689	21.306	1.092	
8 (8)	139.56	0.291	0.150	0.548	312.903	17.594	1.103	
9 (8)	139.56	0.291	0.150	0.548	312.903	17.594	1.065	
10 (8)	133.32	0.291	0.150	0.548	355.539	18.234	1.072	
11 (8)	134.09	0.291	0.150	0.548	316.595	8.603	1.031	
12 (10)	131.02	0.336	0.157	0.637	247.241	12.998	1.049	
13 (10)	127.81	0.336	0.157	0.637	100.052	13.021	1.039	
14 (41)	129.65	0.336	0.157	0.637	317.906	9.844	1.061	

Discussion

Lophelia pertusa morphotypes and the influence of environmental factors

This study reveals the presence of two distinct L. pertusa colony morphologies at MRC (“cauliflower”) and PISCES area (“bush-shaped”). Different morphotypes have already been documented in tropical scleractinian corals, which are known to display several growth forms due to an interplay between factors including: (1) genetic factors (Willis & Ayre, 1985), (2) different influences of abiotic factors such as depth as a proxy for temperature (Barnes, 1973), water movement or turbidity (Foster, 1979; Miller, 1995; Dullo, 2005; Smith, Barshis & Birkeland, 2007) or (3) a combination of genetic and abiotic features (e.g., Via & Lande, 1985). Similar observations concerning CWC colony plasticity have been so far reported by Freiwald (2002), Orejas et al. (2009) and Gori et al. (2013). Unravelling environmental conditions leading to the differentiation in L. pertusa colony shape could lead to the use of morphotypes as a bio-indicator of environmental conditions, an idea initially suggested by Grigg (1972) for gorgonians.

The different morphologies found in the two sites could be due to differences in environmental conditions experienced by the colonies at these locations. MRC and PISCES are respectively constituted by reef structures and coral patches (Wilson, 1979; Roberts et al., 2009). These two sites also display different depth ranges: the reefs studied in the MRC are located between 100–137 m depth whereas the PISCES coral patches were found at a deeper bathymetric range of 230–266 m. Moreover, the MRC site displayed distinct temperature and salinity properties compared to PISCES (Fig. 6), as MRC has a marked coastal influence, and addition of fresh water run-off. Oxygen concentration and Dissolved Inorganic Carbon (DIC) also displayed lower values at PISCES than at MRC as a consequence of the deeper depth range of PISCES (Findlay et al., 2014).

Figure 6 T–S plot showing the potential temperature vs salinity (sal), with isopycnals (grey lines) for all the ROV transects.

Colours represent the different transects (and sites): blue colours show the transects within the Mingulay Reef Complex (MRC) while red colours show the transects within the PISCES areas (PA).

Furthermore, at MRC, the presence of two distinct mechanisms controlling food supply has previously been described (Davies et al., 2009; Duineveld et al., 2012; Findlay et al., 2013). Colonies at MRC receives warmer plankton-rich water thanks to a tidal downwelling of surface water which further circulates to colonies located at BR. Colonies at MR benefit from re-suspended matter from advected deep bottom water (Davies et al., 2009; Duineveld et al., 2012; Findlay et al., 2013).

The oceanographic and food supply regimes at the PISCES site are not yet as well characterised. Particulate Organic Carbon (POC) sampling was carried out during the JC073 Cruise but analysed data is unfortunately not yet available. High POC values have been demonstrated to be an important carbon source for benthic communities including suspension feeding corals and gorgonians (Ribes, Coma & Gili, 1998; Houlbrèque & Ferrier-Pages, 2009; Wagner et al., 2011). Here we could hypothesize that if higher POC values at MRC were measured, POC could explain the higher number of well-developed cauliflower colonies present in MRC compared to PISCES site. Indeed, the MRC site displayed a coral population with colonies at different development degrees (different sizes) whereas the PISCES site displayed homogenous large colonies, perhaps indicative of a more senescent population. This could be better determined with the future addition of POC data.

Colony size, layer thickness estimation and environmental factors

Total colony size and dead layer thickness measured here displayed a higher inter-colony variability than living layer thickness and LL : WC ratio. However, no significant differences could be detected between MRC and PISCES sites. Interestingly living layer thickness was positively correlated to the whole colony size whereas the LL : WC ratio was negatively correlated to the whole colony size (Fig. 5). This could suggest that as a colony develops and its growth rate slows down (as suggested by Brooke & Young, 2009; Lartaud et al., 2013), the proportion of living polyps in the colony decreases. To our knowledge this constitutes the first quantitative analysis of the layering displayed by a reef framework forming CWC. However, the pattern observed here is similar to those described in the facultative zooxanthellate scleractinian coral Oculina varicosa (Reed, 2002). In Reed’s work it was argued that the death of deep-dwelling azooxanthellate coral tissue was due to limited water flow in the core of these colonies because of the dense branching network (Reed, 2002). Flow intensity has been previously demonstrated to influence capture rate of tropical corals (McFadden, 1986; Helmuth & Sebens, 1993; Johnson & Sebens, 1993; Helmuth, Sebens & Daniel, 1997; Sebens et al., 1998; Hoogenboom, Connolly & Anthony, 2008). In L. pertusa, water velocity is also known to impact capture efficiency (Purser et al., 2010; Orejas et al., 2016). However, no significant correlation between abiotic factors including average current speed and layer sizes could here be found to explain the reasons for the differences detected in the colonies at MRC. The 3D hydrodynamic model used to extract abiotic data (Moreno-Navas et al., 2014) has a 100 m spatial resolution, which is most probably too coarse to reveal the impact of the hydrodynamic on a single colony development.

It is important to take into account that the LL : WC ratio as calculated here never exceeded 0.27, meaning that the living layer never represents more than one quarter of the whole colony size. Initial short-term laboratory experiments show that L. pertusa seemed resilient to change in water chemistry and lower pH conditions (Maier et al., 2013; Hennige et al., 2014b). However, a recent study published by Hennige et al. (2015) shows that even though L. pertusa is able to physiologically adapt to OA conditions over a longer time period, its skeleton becomes significantly weaker, leading to breakage of the framework and higher susceptibility to bioerosion and mechanical damage. As the dead layer constitutes the great majority of a L. pertusa colony as shown by our measurements, the consequences of OA on CWC reefs and the habitats they support, could be worse than expected. A negative influence of low pH levels on the dead layer of the colony would lead to a weakening of the skeleton and its breakage, with the consequent destruction of the three-dimensional structure of the colonies and the reef. Further investigation of colony layering and morphotypes at other reefs, as well as growth rate in situ validations, is needed to further investigate the use of LL : WC ratio measurements as a bio-indicator of colony health and susceptibility to OA.

Colony health indicators are vital in order to develop robust protocols to monitor changes in CWC colony health over time, particularly to assess the effectiveness of deep-water Marine Protected Areas (MPA) created for their long-term conservation. Recent work has shown the slow growth rate and low recovery potential of deep-water coral habitats at the Darwin Mounds at 1,000 m water depth in the Rockall Trough (Huvenne et al., 2016). Our work demonstrates a simple metric that could be scaled up using already collected ROV footage or autonomous survey approaches (see Wynn et al., 2014) to characterise larger habitat areas and gather datasets from many more coral colonies than possible in the present study. When combined with machine-learning image analysis and a robust understanding of deep-water MPA network connectivity (Fox et al., 2016) these approaches will greatly improve our understanding of deep-sea habitats and our ability to monitor them over time.

Supplemental Information

Supplemental Information 1 Table of the Lophelia pertusa colony measurements used in this study

Click here for additional data file.

Supplemental Information 2 Still Image of colony 1

Click here for additional data file.

Supplemental Information 3 Still Image of colony 2

Click here for additional data file.

Supplemental Information 4 Still Image of colony 3

Click here for additional data file.

Supplemental Information 5 Still Image of colony 4

Click here for additional data file.

Supplemental Information 6 Still image of colony 5

Click here for additional data file.

Supplemental Information 7 Still Image of colony 6

Click here for additional data file.

Supplemental Information 8 Still image of colony 7

Click here for additional data file.

Supplemental Information 9 Still Image of colony 8

Click here for additional data file.

Supplemental Information 10 Still image of colony 9

Click here for additional data file.

Supplemental Information 11 Still Image of colony 10

Click here for additional data file.

Supplemental Information 12 Still Image of colony 11

Click here for additional data file.

Supplemental Information 13 Still image of colony 12

Click here for additional data file.

Supplemental Information 14 Still image of colony 13

Click here for additional data file.

Supplemental Information 15 Still image of colony 14

Click here for additional data file.

Supplemental Information 16 Still Images of colony 15

Click here for additional data file.

Supplemental Information 17 Still Image of colony 16

Click here for additional data file.

Supplemental Information 18 Still Image of colony 17

Click here for additional data file.

Supplemental Information 19 Still Image of colony 18

Click here for additional data file.

JV, JMN and JMR acknowledge additional support from University of Edinburgh Changing Ocean group. JV and CO acknowledge support from the IEO. We thank the captain, crew and scientific participants of RRS James Cook cruise 073 for assistance at sea.

Additional Information and Declarations

Competing Interests

Author Contributions

Data Availability

Helen S. Findlay is an employee of Plymouth Marine Laboratory, Plymouth, UK.

Johanne Vad conceived and designed the experiments, performed the experiments, analyzed the data, wrote the paper, prepared figures and/or tables, reviewed drafts of the paper.

Covadonga Orejas conceived and designed the experiments, wrote the paper, prepared figures and/or tables, reviewed drafts of the paper.

Juan Moreno-Navas and Helen S. Findlay contributed reagents/materials/analysis tools, prepared figures and/or tables, reviewed drafts of the paper.

J. Murray Roberts conceived and designed the experiments, wrote the paper, reviewed drafts of the paper.

The following information was supplied regarding data availability:

The raw data has been supplied as a Supplementary File.

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
