# Peer review of "Assessing the living and dead proportions of cold-water coral colonies: implications for deep-water Marine Protected Area monitoring in a changing ocean"

_PeerJ, doi:10.7717/peerj.3705_

## Round 0.1 · original submission · Major Revisions

This study provides very valuable data on little known habitats, as highlighted by both reviewers.

There are some major concerns raised by Reviewer 1, however, regarding the approach used to measure coral colony size. A more comprehensive and detailed explanation of how the measures were made and standarised needs to be provided before the study can be considered for publication.

In addition to the other points raised by both Reviewer 1 and 2, a thorough revision of the text is needed to improve the flow and overall clarity of the MS.

Reviewer 1 ·

Basic reporting

English is not clear in some cases, I have annotated the pdf to highlight ambiguous statements and in some cases I commented or provided suggestions of how I think the English can be improved. Yet I did not revise the entire manuscript thoroughly as I believe the authors should do this before submitting a manuscript for revision.
Also the manuscript uses many acronyms and it is difficult to keep track of all of them – I suggest authors should consider spelling out as many terms as possible rather than using acronyms.Other minor suggestions are noted on the pdf.

My expertise lies with tropical corals, remote sensing and climate change, not cold water corals. I am not aware of any important references that were missed, but the manuscript seems to have many self-citations. I advise another reviewer with cold water coral expertise comments on this point.

The structure of the manuscript is good, However, several figures are sub standard they need significant improvement – please see specific comments on pdf

This is a self-contained manuscript – but hypothesis are not explicitly explained.

Experimental design

Original primary research within Aims and Scope of the journal.
Yes

Research question well defined, relevant & meaningful. It is stated how research fills an identified knowledge gap. The submission should clearly define the research question, which must be relevant and meaningful. The knowledge gap being investigated should be identified, and statements should be made as to how the study contributes to filling that gap.
Yes

Rigorous investigation performed to a high technical & ethical standard. The investigation must have been conducted rigorously and to a high technical standard. The research must have been conducted in conformity with the prevailing ethical standards in the field.

No – my biggest concern about this manuscript is that it is not possible to determine the precision of the measurements made. Based on both the text and the figures provided it seems to me that the lines used in Image J to measure coral size could have stopped anywhere, they do not seem to stop at the top of the coral and from previous experience with similar measurements I know it can be very tricky to do this in a standardized fashion.
Unless more detail and exact method of standardization of these measurements is provided I cannot recommend publication.

Methods described with sufficient detail & information to replicate. Methods should be described with sufficient information to be reproducible by another investigator.

See comment above

Validity of the findings

Impact and novelty not assessed. Negative/inconclusive results accepted. Meaningful replication encouraged where rationale & benefit to literature is clearly stated. Decisions are not made based on any subjective determination of impact, degree of advance, novelty, being of interest to only a niche audience, etc. Replication experiments are encouraged (provided the rationale for the replication, and how it adds value to the literature, is clearly described); however, we do not allow the ‘pointless’ repetition of well known, widely accepted results.

No, see comment under experimental design and validity of measurements.

Data is robust, statistically sound, & controlled. The data on which the conclusions are based must be provided or made available in an acceptable discipline-specific repository. The data should be robust, statistically sound, and controlled.

No, see comment under experimental design and validity of measurements.

Also, images for all 18 colonies and measurements should be provided in supplementary material – not just the table of derived values. Given the rarity of these ecosystems and data, it is essential to provide the imagery.
Conclusion are well stated, linked to original research question & limited to supporting results. The conclusions should be appropriately stated, should be connected to the original question investigated, and should be limited to those supported by the results.

NA – I did not finish reading the manuscript as I don’t think it has merit unless details are
provided regarding the methodology and measurements.

Speculation is welcome, but should be identified as such.

NA – I did not finish reading the manuscript as I don’t think it has merit unless details are provided regarding the methodology and measurements.

Additional comments

This manuscript investigates the live vs dead coral tissue of 18 coral colonies in deep cold water reefs. Cold water corals are not studied enough and it is important to improve our understanding of these systems and their related organisms.

My biggest concern about this manuscript is that it is not possible to determine the precision of the measurements made. Based on both the text and the figures provided it seems to me that the lines used in Image J to measure coral size could have stopped anywhere, they do not seem to stop at the top of the coral and from previous experience with similar measurements I know it can be very tricky to do this in a standardized fashion.

Unless more detail and exact method of standardization of these measurements is provided I cannot recommend publication.

The manuscript also needs significant improvement on its English grammar, at present it is not very clear and in some instances it is difficult to understand what the authors mean. I recommend authors ask a good scientific writer (preferably whose first language is English) to revise their manuscript prior to resubmission. I have made detailed comments on the pdf.

The manuscript figures are of low quality and need to be significantly improved as well. I have made detailed comments on the pdf.

Annotated reviews are not available for download in order to protect the identity of reviewers who chose to remain anonymous.

·

Basic reporting

The structure of the paper meets required standards. The language is clear and the literature cited was fairly extensive given the relatively short paper. Figures etc were clear and of good quality.

The discussion is rather disjointed in places with statements that do not seem to have a context. A little more work is needed on the discussion

The addition of a hypothesis would improve the paper. Since the authors are comparing two different sites, it would be relatively easy to formulate a hypothesis about expected differences between the sites.

Experimental design

The survey design was unbalanced, which reduced the comparative value of the study but did not invalidate it.

The study is focused on the application of a new method for assessing coral growth, and as such is useful.

I think there needs to be more detail on the abiotic factors: what the values were and how they were derived from the model. They are superficially described but the reader has no information with which to assess the colony measurements or the discussion.

The absence of temperature as an abiotic correlate is surprising as it is one of the more important factors that influence growth. Temperature should be included in the analysis, or if the data is not available, it should be considered in the discussion.

There is no abiotic data for the Pisces sites, which is a flaw as it does not allow site comparisons. If this data is not available, the effects on the analysis should be acknowldged in the discussion

Validity of the findings

Additional information the aboitic factors is needed to determine whether the data used is robust and statistically sound. The statistical results for these analyses should be presented fully - a table would suffice.

Additional comments

A valuable paper but needs some work on the discussion and data analysis. I have made some additional comments and suggested edits in the attached PDF file.

---

## Round 0.2 · accepted · Accept

While in production, please review the grammar in the legend of the new Figures 2c and 3c - the text is currently difficult to understand.

·

Basic reporting

This is a second review and the authors have addressed all my initial concerns appropriately. I recommend accepting the re-submitted version of this manuscript.

Experimental design

No comment

Validity of the findings

No comment

Additional comments

No comment